# Coronavirus surveillance in wildlife from two Congo basin countries detects RNA of multiple species circulating in bats and rodents

Charles Kumakamba[1], Fabien R. Niama[2], Francisca Muyembe[1], Jean-Vivien Mombouli[2], Placide Mbala Kingebeni[1], Rock Aime Nina[3], Ipos Ngay Lukusa[1], Gerard Bounga[4], Frida N'Kawa[1], Cynthia Goma Nkoua[2], Joseph Atibu Losoma[1], Prime Mulembakani[1], Maria Makuwa[1,5], Ubald Tamufe[6], Amethyst Gillis[7¤a], Matthew LeBreton[8], Sarah H. Olson[4], Kenneth Cameron[4¤b], Patricia Reed[4], Alain Ondzie[4], Alex Tremeau-Bravard[9], Brett R. Smith[9], Jasmine Pante[9], Bradley S. Schneider[7¤c¤d], David J. McIver[10¤e], James A. Ayukekbong[10¤f], Nicole A. Hoff[11], Anne W. Rimoin[11], Anne Laudisoit[12], Corina Monagin[7,9], Tracey Goldstein[9], Damien O. Joly[4,10¤g], Karen Saylors[5,7], Nathan D. Wolfe[7], Edward M. Rubin[7], Romain Bagamboula MPassi[13], Jean J. Muyembe Tamfum[14], Christian E. Lange[5,10]*

1 Metabiota Inc, Kinshasa, Kinshasa, Democratic Republic of the Congo, 2 National Laboratory of Public Health, Brazzaville, Republic of the Congo, 3 Ministry of Agriculture and Livestock, Brazzaville, Republic of the Congo, 4 Wildlife Conversation Society, Bronx, New York, United States of America, 5 Labyringth Global Health St. Petersburg, Florida, United States of America, 6 Metabiota Cameroon Ltd, Yaoundé, Centre, Cameroon, 7 Metabiota Inc, San Francisco, California, United States of America, 8 Mosaic, Yaoundé, Centre, Cameroon, 9 One Health Institute, School of Veterinary Medicine, University of California, Davis, California, United States of America, 10 Metabiota Inc, Nanaimo, British Columbia, Canada, 11 Fielding School of Public Health, University of California, Los Angeles, California, United States of America, 12 EcoHealth Alliance, New York, New York, United States of America, 13 Ministry of National Defense, Brazzaville, Republic of Congo, 14 Institut National de Recherche Biomédicale, Kinshasa, Kinshasa, Democratic Republic of the Congo

¤a Current address: Development Alternatives, Inc., Washington DC, United States of America
¤b Current address: Unites States Fish and Wildlife Service, Bailey's Crossroads, Virginia, United States of America
¤c Current address: Etiologic, Oakland, California, United States of America
¤d Current address: Pinpoint Science, San Francisco, California, United States of America
¤e Current address: Institute for Global Health Sciences, University of California, San Francisco, California, United States of America
¤f Current address: Southbridge Care, Cambridge, Ontario, Canada
¤g Current address: British Columbia Ministry of Environment and Climate Change Strategy, Victoria, British Columbia, Canada
* clange_virology@gmx.de

**Data Availability Statement:** All relevant data are within the paper and its Supporting Information files.

## Abstract

Coronaviruses play an important role as pathogens of humans and animals, and the emergence of epidemics like SARS, MERS and COVID-19 is closely linked to zoonotic transmission events primarily from wild animals. Bats have been found to be an important source of coronaviruses with some of them having the potential to infect humans, with other animals serving as intermediate or alternate hosts or reservoirs. Host diversity may be an important contributor to viral diversity and thus the potential for zoonotic events. To date, limited research has been done in Africa on this topic, in particular in the Congo Basin despite

**Funding:** The PREDICT Consortium (https://ohi.vetmed.ucdavis.edu/programs-projects/predictproject/authorship) received awards GHN-A-OO-09-00010-00 and AID-OAA-A-14-00102 from the United States Agency for International Development (https://www.usaid.gov). The contributions to/work on this manuscript of all listed authors were/was funded exclusively through these awards. The funders had no role in study design, data collection and analysis, decision to publish, or preparation of the manuscript.

**Competing interests:** Metabiota (https://metabiota.com), Labyrinth Global Health (https://www.labyrinthgh.com), and Mosaic (https://mosaic.cm) are private contractors who receive funds from international donor organizations to conduct technical assistance and research. Metabiota employs or employed CK, FM, PMK, INL, FNK, JAL, PM, MM, UT, AG, BSS, DJM, JAA, CM, DOJ, KS, NDW, EMR, and CEL, Mosaic employs ML, and Labyrinth Global Health employs KS, MM and CEL. There are no patents, products in development or marketed products associated with this research to declare. The association with commercial entities that some of the authors have does not alter our adherence to PLOS ONE policies on sharing data and materials.

frequent contact between humans and wildlife in this region. We sampled and, using consensus coronavirus PCR-primers, tested 3,561 wild animals for coronavirus RNA. The focus was on bats (38%), rodents (38%), and primates (23%) that posed an elevated risk for contact with people, and we found coronavirus RNA in 121 animals, of which all but two were bats. Depending on the taxonomic family, bats were significantly more likely to be coronavirus RNA-positive when sampled either in the wet (*Pteropodidae* and *Rhinolophidae*) or dry season (*Hipposideridae*, *Miniopteridae*, *Molossidae*, and *Vespertilionidae*). The detected RNA sequences correspond to 15 alpha- and 6 betacoronaviruses, with some of them being very similar (>95% nucleotide identities) to known coronaviruses and others being more unique and potentially representing novel viruses. In seven of the bats, we detected RNA most closely related to sequences of the human common cold coronaviruses 229E or NL63 (>80% nucleotide identities). The findings highlight the potential for coronavirus spillover, especially in regions with a high diversity of bats and close human contact, and reinforces the need for ongoing surveillance.

## Introduction

Coronaviruses are relatively large enveloped viruses with a single-stranded positive-sense RNA genome of 26–32 kilobases that form their own taxonomic family within the *Nidovirales* order of viruses [1]. There are two *Coronaviridae* subfamilies, *Letovirinae* and *Orthocoronavirinae*, and the latter contains the genera *Alpha-* and *Betacoronavirus*, with viruses infecting mammalian species as well as the genera *Gamma-* and *Deltacoronavirus* that primarily contain viruses found in birds [2]. Although known for decades as important enteric and respiratory pathogens in domestic animals, and as causative agent of mild respiratory infections in humans, it was only the emergence of severe acute respiratory syndrome coronavirus (SARS-CoV) in humans in 2002 that brought coronaviruses broader attention [3]. The emergence and sporadic re-emergence of Middle East respiratory syndrome coronavirus (MERS-CoV) since 2012 and the global COVID-19 pandemic caused by SARS-CoV-2 have highlighted the enormous importance of this viral family in the context of global public health [4–6].

Coronaviruses identical or closely related to SARS-CoV-1, MERS-CoV and SARS-CoV-2 have been found in civets, camels, and bats, supporting zoonotic events as the most likely source of the respective outbreaks in humans [5, 7–11]. Studies to identify the origin of these zoonotic viruses also led to the discovery of many other, related or completely novel, animal coronaviruses in the process; in particular they have detected an astonishing diversity of alpha- and betacoronaviruses in bats. Bats (*Chiroptera*) are the second most diverse order of mammals second only to rodents and with its two suborders *Yinpterochiroptera* and *Yangochiroptera* represent approximately 20% of all known mammalian species [12].

Coronaviruses detected in bats include relatives of coronaviruses previously identified in other hosts, which led to the hypothesis that bats are a reservoir for coronaviruses and that these viruses are crossing into other non-bat species on a somewhat regular basis [13–16]. As a result, they may establish a novel permanent virus-host relationship, as in the case of MERS-CoV and camels, or a transient relationship as in the case of SARS-CoV-1 and civets. However most interspecies transmissions are likely dead ends for the virus and remain undetected [16–18]. The human common cold viruses HCoV-229E and HCoV-NL63 are most likely animal origin viruses that succeeded in establishing a permanent relationship with humans after crossing species barriers directly or indirectly from bats [13, 14, 19, 20]. Coronaviruses OC43

and HKU-1, which also cause common cold in humans, are likewise expected to have originated in animals, though likely in rodents rather than bats [6, 21].

In sum, there is thus mounting biological evidence that spillover has happened repeatedly in the past, continues to happen today, and will likely continue to occur in the future. Hence it is important to study animal coronaviruses to characterize the risks posed by these potentially emerging viruses, to understand the dynamics of the emergence of these pathogens, and to make informed decisions concerning prevention and risk mitigation [16, 22].

However, the virus' biology is only one piece of the puzzle. We know that the emergence and epidemic spread of SARS-CoV-1 and likely SARS-CoV-2 are linked to human behavioral factors, such as close contact with wild animals, and with factors such as biodiversity and wildlife abundance, important prerequisites for virus diversity. Hotspots for zoonotic disease emergence generally exist where humans are actively encroaching on such animal habitats [23, 24], as is happening in Southeast Asia and Central Africa. While potential sources of zoonotic coronaviruses are increasingly being explored, a great deal remains to be documented in most parts of the biodiverse African continent, especially in Central Africa. Findings from countries such as Gabon, Kenya, Madagascar, Rwanda, South Africa and others suggest there are many coronaviruses circulating, primarily in bats, including species related to pathogens such as SARS-CoV-1, MERS-CoV, HCoV-229E and HCoV-NL63 [16, 25–32].

In the Democratic Republic of the Congo (DRC) and the Republic of Congo (ROC) contact with wildlife is common for large parts of the population via the value chain (food or otherwise), as pests in house and fields, at peri-domestic and co-feeding interfaces, or in the context of conservation and tourism [33, 34]. This close contact does not only involve risks for humans, but also potentially for endangered animal species such as great apes [35]. To explore the coronavirus presence in wildlife in this region representing one of the most biodiverse places on the African continent, we launched large scale sampling of primarily bats, rodents and non-human primates (NHPs). Our goal was to determine the diversity of coronaviruses circulating, using a consensus Polymerase Chain reaction (PCR) approach, coupled with the collection of, and coupling with, ecological data.

## Materials and methods

Sample acquisition at multiple locations in DRC and ROC took place between 2006 and 2018 and differed depending on the species and interface (Fig 1, S1 Fig, S1 Table). Animals in peri-domestic settings were captured and released after sampling (bats, rodents and shrews only), while samples from the (bushmeat) value chain were collected from freshly killed animals voluntarily provided by local hunters upon their return to the village following hunting, or by vendors at markets. Fecal samples were collected from free-ranging NHPs [36]. Some NHP samples were also collected during routine veterinary exams in zoos and wildlife sanctuaries. Hunters and vendors were not compensated, to avoid incentivizing hunting. Sampling of animals was conducted based on specifically identified "sites", where it was determined that there was considerable interaction between humans and wildlife, with this interaction ranging from hunting, to consumption, to interactions with animals as pests. Across both DRC and ROC, sites were visited at varying frequencies, based on ease of access to sites, return on effort (number of animals sampled for amount of effort expended in collection), and seasonal considerations. Efforts were made to visit each site at least twice each year, corresponding to a sampling event in both the wet and dry seasons. While repeat visits were made to most sites, no repeat sampling of individual animals occurred for bushmeat (as they are used or consumed quickly). It is possible that live animals which were captured and released were sampled more than once, as no tags or identifying methods were used, though repeat sampling is

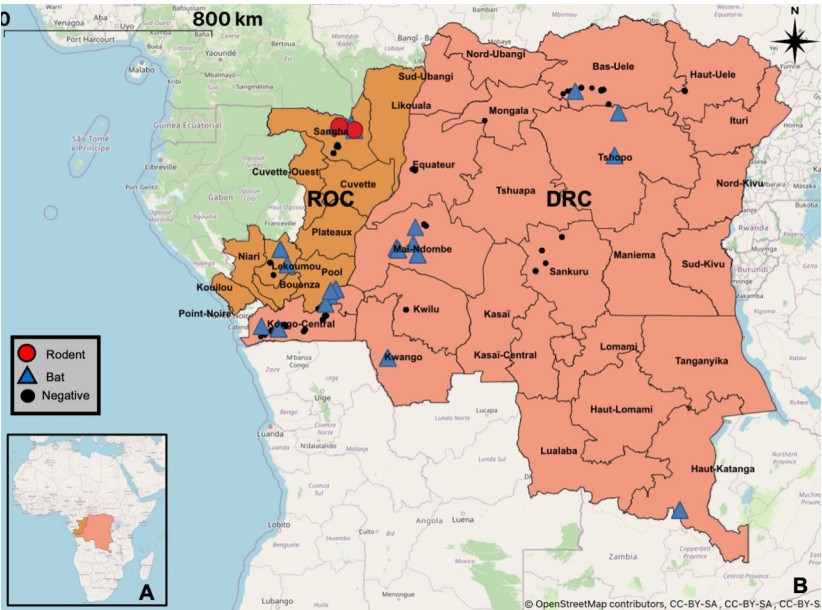

**Fig 1. Sampling sites map.** Geographical map indicating all sampling sites within the Republic of Congo and the Democratic Republic of the Congo. Locations where coronaviruses were detected are highlighted with blue triangles for bats and red circles for rodents. Sampling sites without viral RNA detection are marked by black dots (see also S1 Fig). Base map and data from OpenStreetMap and OpenStreetMap Foundation.

unlikely given the population of the collected species, individual life span, and the relatively small number of individuals captured. Identification was done in the field by trained field ecologists as well as retrospectively based on various field guides and other resources including those by including Kingdon and Monadjem [37–39]. Sample collection focused on oral/respiratory and intestinal/fecal organ systems and transmission routes, but other samples were collected when available and tested alongside, to detect possible infections of other organ systems.

Animal capture and specimen collection was approved by the Institutional Animal Care and Use Committee (UCDavis IACUC, Protocol #s 16048, 16067, 17803, and 19300), the Institute Congolais pour la Conservation de la Nature (0374/ICCN/DG/ADG/ADG/KV/2011) in DRC, and the Ministry of Forest Economy and Sustainable Development (1102/MEFDD/DGEFDFAP-SPR) and the Ministry of Scientific Research and Technical Innovation (permit number 014/MRS/DGRST/DMAST, 018/MRSIT/DGRST/DMAST) in the RoC.

Oral and rectal swab samples were collected into individual 2.0 ml screw-top cryotubes containing 1.5 ml of either Universal Viral Transport Medium (BD), RNA later, lysis buffer, or Trizol® (Invitrogen), while pea-sized tissue samples were placed in 1.5ml screw-top cryotubes containing 500ul of either RNA later or lysis buffer (Qiagen), or without medium. All samples were stored in liquid nitrogen as soon as practical. Sample collection staff wore dedicated clothing: N95 masks, nitrile gloves, and protective eyewear during animal capture, handling and sampling.

RNA was extracted either manually using Trizol®, with an Qiagen AllPrep kit (tissue), Qiagen Viral RNA Mini Kit (swabs collected prior to 2014), or with a Zymo Direct-zol RNA kit (swabs collected after 2014) and stored at -80˚C. Afterwards RNA was converted into cDNA using a Maxima H Minus First Strand cDNA Synthesis Kit (Thermo Scientific) or GoScript™ Reverse Transcription kit (Promega) and stored at -20˚C until analysis. Two conventional nested broad range PCR assays, both targeting conserved regions within the

RNA-Dependent RNA Polymerase gene (RdRp) were used to test the samples for coronavirus RNA. The first PCR as published by Quan et al. amplifies a product of approximately 286nt between the primer binding sites and was used as published. The first round (CoV-FWD1: CGT TGG IAC WAA YBT VCC WYT ICA RBT RGG and CoV-RVS1: GGT CAT KAT AGC RTC AVM ASW WGC NAC ATG) and second round (CoV-FWD2: GGC WCC WCC HGG NGA RCA ATT and CoV-RVS2: GGW AWC CCC AYT GYT GWA YRT C) primers of this PCR were specifically designed for the detection of a broad range of coronaviruses [40]. The second PCR as published by Watanabe et al. was used in two modified versions: one of them specifically targeting a broad range of coronaviruses in bats, the second one broadly targeting coronaviruses of other hosts [41]. In both cases, the first round of the semi nested PCR utilized the primers CoV-FWD3 (GGT TGG GAY TAY CCH AAR TGT GA) and CoV-RVS3 (CCA TCA TCA SWY RAA TCA TCA TA). In the second round, either CoV-FWD4/Bat (GAY TAY CCH AAR TGT GAY AGA GC) or CoV-FWD4/Other (GAY TAY CCH AAR TGT GAU MGW GC) were used as forward primers, while the reverse primer was again CoV-RVS3. Both versions amplify 387nt between the primer binding sites. A plasmid with binding sites for both the Quan and the Watanabe assays but otherwise lacking coronavirus sequence was used as positive control.

PCR products were subjected to gel electrophoresis on a 1.5% agarose gel and products of the expected amplicon sizes were excised. DNA was extracted using the Qiagen QIAquick Gel Extraction Kit and either sequenced by Sanger sequencing at the UC Davis DNA sequencing facility or was sent for commercial Sanger sequencing (GATC or Macrogen). Extracts with low DNA concentrations were cloned prior to sequencing. All results from sequencing were analyzed in the Geneious 7.1 software, and primer trimmed consensus sequences compared to the GenBank database (BLAST N, NCBI).

Viral sequences were deposited in the GenBank database under submission numbers KX284927-KX284930, KX285070-KX285095, KX285097-KX285105, KX285499-KX285513, KX286248-KX286258, KX286264-KX286286, KX286295-KX286296, KX286298-KX286322, MT064119-MT064126, MT064226, MT064272, MT081973, MT081997-MT082004, MT082032, MT082059-MT082060, MT082072, MT082123-MT082136, MT082145, MT082299, MT222036- MT222037.

Maximum likelihood phylogenetic trees were constructed including different genera (Alpha, Beta and Gamma) and species of known coronaviruses, as well as species/sub-species detected in DRC and ROC during the PREDICT project. Only a single sequence was included representing sequences with nucleotide identities of more than 95%. Multiple sequence alignments were made in Geneious (version 11.1.3, ClustalW Alignment). Bayesian phylogeny of the polymerase gene fragment was inferred using MrBayes (version 3.2) with the following parameters: Datatype = DNA, Nucmodel = 4by4, Nst = 1, Coavion = No, # States = 4, Rates = Equal, 2 runs, 4 chains of 5,000,000 generations. The sequence of an avian Gamma Coronavirus (NC_001451) served as outgroup to root the trees, and trees were sampled after every 1,000 steps during the process to monitor phylogenetic convergence [42]. The average standard deviation of split frequencies was below 0.006 for the Watanabe PCR amplicon based analysis and below 0.0029 for the Quan PCR amplicon based analysis (MrBayes recommended final average <0.01). The first 10% of the trees were discarded and the remaining ones combined using TreeAnnotator (version 2.5.1; http://beast.bio.ed.ac.uk) and displayed with FIG-TREE (1.4.4; http://tree.bio.ed.ac.uk/) [43].

Ecological data related to the locality and the host animals was compiled and analyzed with respect to a correlation with the frequency of virus detection. The data included sex, human interface at which the animals were collected and sampled (value chain or other), and local calendric season (wet/dry), and were evaluated using two-tailed Chi-square tests with Yates

correction ($C^2Y$). Season for each sampling site was based on the next closest location represented in the climate-data.org data set, and defined as dry for months with average rainfall below 100mm/month and as wet for months above.

## Results

A total of 3,554 animals (2,630 from DRC and 931 from RoC) were sampled and tested, of which 1,356 were bats (24 genera), 1,347 rodents (33 genera), 829 NHPs (14 genera), and 22 shrews, (Fig 1, S1 Fig, S1 Table). The majority of the 5,579 collected samples were oral (2,258) or rectal (2,238) swabs, with others being tissue samples, including liver and spleen (385), lung (184) or intestinal tract (175), as well as feces (160), blood, serum or plasma (140), kidney (24) and brain samples (4). Coronavirus RNA was detected in one or more samples from 121 animals, with 102 of the coronavirus RNA positive samples being rectal and 23 being oral swabs as well as one pooled liver and spleen sample. The CoV positive rate in intestinal/fecal samples (rectal swabs, intestinal tissue, feces) was with 3.96% significantly higher (N<0.0001 C2Y) than in oral/respiratory samples (oral swabs, lung tissue) with 0.94% or blood/immune system samples (liver, spleen, blood, serum or plasma) with 0.19%. The difference between the latter two was not significant. Only a single coronavirus (n = 1) was solely detected in an oral swab, in all other cases detection in rectal swabs exceeded detection in oral swabs by at least factor of two.

Viral RNA in afore mentioned 121 animals was amplified with either both PCRs (n = 33), the Watanabe PCR assay only (n = 48) or the Quan PCR assay only (n = 40) (S2 Table). The Watanabe assay amplified alphacoronavirus RNA in 13 and betacoronavirus RNA in 70 cases, of which 9 and 4 respectively differed by more than 5% from each other, while the Quan assay amplified alphacoronavirus RNA in 17 and betacoronavirus RNA in 52 cases, of which 8 and 5 respectively differed by more than 5% from each other. The difference between the tests with regards to alpha- and betacoronavirus RNA detection were not statistically significant.

Two of the animals with detected coronavirus RNA were rodents (<1% of sampled rodents), while 119 were bats (8.8% of sampled bats). Coronavirus RNA positive animals were found in 25% (27/106) of bat sampling events (same location and same day) and <1% (2/235) of rodent sampling events (S1 and S2 Tables, S2 Fig). In 10 of the bat sampling events, a single coronavirus RNA positive bat was among the tested animals, while in 17 events the number of bats positive for coronavirus ranged from 2 to 16 (S2 Table). RNA was detected in two species of rodents, one *Deomys ferrugineus* (1/1) and one *Malacomys longipes* (1/38), and in at least 14 different bat species, namely *Chaerephon pumilus* (4/62), *Chaerephon sp.* (2/6), *Eidolon helvum* (23/103), *Epomops franqueti* (22/146), *Hipposideros caffer* (1/5), *Hipposideros gigas* (1/2), *Hipposideros ruber* (3/21), *Hipposideros sp.* (1/4), *Megaloglossus woermanni* (11/118), *Micropteropus pusillus* (29/417), *Miniopterus inflatus* (3/6), *Mops condylurus* (8/105), *Myonycteris sp.* (4/4), *Rhinolophus sp.* (1/62), *Scotophilus dinganii* (1/29) and *Triaenops persicus* (5/26) (Table 1, S2 Table). Among the five bat species from which more than 100 individuals were sampled and tested, *Eidolon helvum* had the highest rate of coronavirus RNA positives (22.3%), followed by *Epomops franqueti* (15.8%), *Megaloglossus woermanni* (8.5%), *Mops condylurus* (7.6%), and *Micropteropus pusillus* (7%) (Table 1). With 10.2% *Yinpterochiroptera* bats had a significantly (N = 0.015 $C^2Y$) higher rate of coronavirus RNA positive animals than *Yangochiroptera* bats with 5.0% (Table 1). No coronavirus RNA positive animals were detected among the sampled NHPs or shrews.

Significant seasonal differences for the rate of coronavirus RNA positive animals were detected across the bats with a 10.5% PCR positive rate in the wet season and a 6.6% rate in the dry season (p = 0.0176) (Table 1, S2 Fig). Bats that were associated with the (bushmeat) value

**Table 1. PCR results by species and season (bats).**

| Suborder, family and species (>10 sampled individuals) | Wet Season | Dry Season | Total |
|---|---|---|---|
| | PCR positives | PCR positives | PCR positives |
| *Yinpterochiroptera* total** | **13.3% (78/586)** | **5.6% (23/408)** | **10.2% (101/994)** |
| *Pteropodidae* total** | **13.6% (77/567)** | **4.0% (12/303)** | **10.2% (89/870)** |
| *Micropteropus pusillus*** | 10.3% (27/263) | 1.3% (2/153) | 7% (29/416) |
| *Epomops franqueti* | 16.5% (18/109) | 13.5% (5/37) | 15.8% (23/146) |
| *Megaloglossus woermanni* | 11.9% (5/42) | 6.6% (5/76) | 8.5% (10/118) |
| *Eidolon helvum* | 22.3% (23/103) | - (0/0) | 22.3% (23/103) |
| *Myonycteris torquata* | 0% (0/11) | 0% (0/11) | 0% (0/22) |
| *Rhinolophidae* total** | **100% (1/1)** | **0% (0/61)** | **1.6% (1/62)** |
| *Hipposideridae* total* | **0% (0/18)** | **25% (11/44)** | **17.7% (11/62)** |
| *Hipposideros ruber* | 0% (0/12) | 33.3% (3/9) | 14.3% (3/21) |
| *Triaenops persicus* | - (0/0) | 13.8% (4/29) | 13.8% (4/29) |
| *Yangochiroptera* total** | **0.6% (1/167)** | **8.7% (17/194)** | **5.0% (18/361)** |
| *Miniopteridae* total | **0% (0/1)** | **20% (3/15)** | **18.8% (3/16)** |
| *Pipistrellus nanus* | 0% (0/1) | 0% (0/10) | 0% (0/11) |
| *Molossidae* total** | **1.1% (1/92)** | **14.8% (13/88)** | **7.8% (14/180)** |
| *Chaerephon pumilus* | 0% (0/33) | 12.5% (4/32) | 6.2% (4/65) |
| *Mops condylurus* | 1.9% (1/52) | 13.2% (7/53) | 7.6% (8/105) |
| *Vespertilionidae* total | **0% (0/74)** | **1.1% (1/91)** | **0.6% (1/165)** |
| *Scotophilus dinganii* | 0% (0/18) | 9% (1/11) | 3.4% (1/29) |
| **Total**** | **10.5% (79/754)** | **6.6% (40/602)** | **8.8% (119/1356)** |

* Significant difference between calendric seasons P<0.05 (Chi-square with Yates correction)

** Highly significant difference between calendric seasons P<0.01 (Chi-square with Yates correction)

chain were more frequently positive for coronavirus RNA (25.4%) than bats sampled at other human animal (peri-domestic) interfaces (5%), and this difference was highly significant (p < 0.0001). Male bats were significantly overrepresented among the Coronavirus RNA positives p = 0.0183), while there was insufficient data to analyze an influence of age (S1 Table).

Upon phylogenetic analysis, the sequences fall into 13 separate clusters based on the Quan PCR amplicon and 13 separate clusters based on the Watanabe PCR amplicon. Based on amplicons obtained with both PCRs from the same sample or animal, the respective Quan and Watanabe sequence clusters Alpha 5, 6, and 7 (Q7 = W2), as well as Beta 1, 2, and 3 correspond to each other. In one bat, RNA corresponding to two different alphacoronaviruses was detected in the oral and the rectal sample by the same PCR assay (ZB12030), while in another bat one PCR assay amplified RNA indicating an alpha- and the other assay an RNA indicating a betacoronavirus (GVF-RC-1006) (S2 Table). Given the overall results, RNA of 15 different alpha- and 6 betacoronaviruses was detected in the study population. In 22 of the sampling events, only a single type/strain of these coronaviruses was detected, two in two events, and three, five or eight in one event each. Identical or very similar coronavirus sequences were found with a spatial distance of up to 1975 km apart and a temporal distance of up to 1708 days (S3 Table).

Although the two coronavirus sequences we detected in rodents were clustering with known sequences from bat alphacoronaviruses, there were no sequences in GenBank that shared more than 80% identities with either of them (Fig 2, S2 Table). The detected bat coronavirus sequences on the contrary mostly clustered closely with known ones, that to a large part were detected in hosts from the same genus (Figs 2 and 3). The majority of the detected

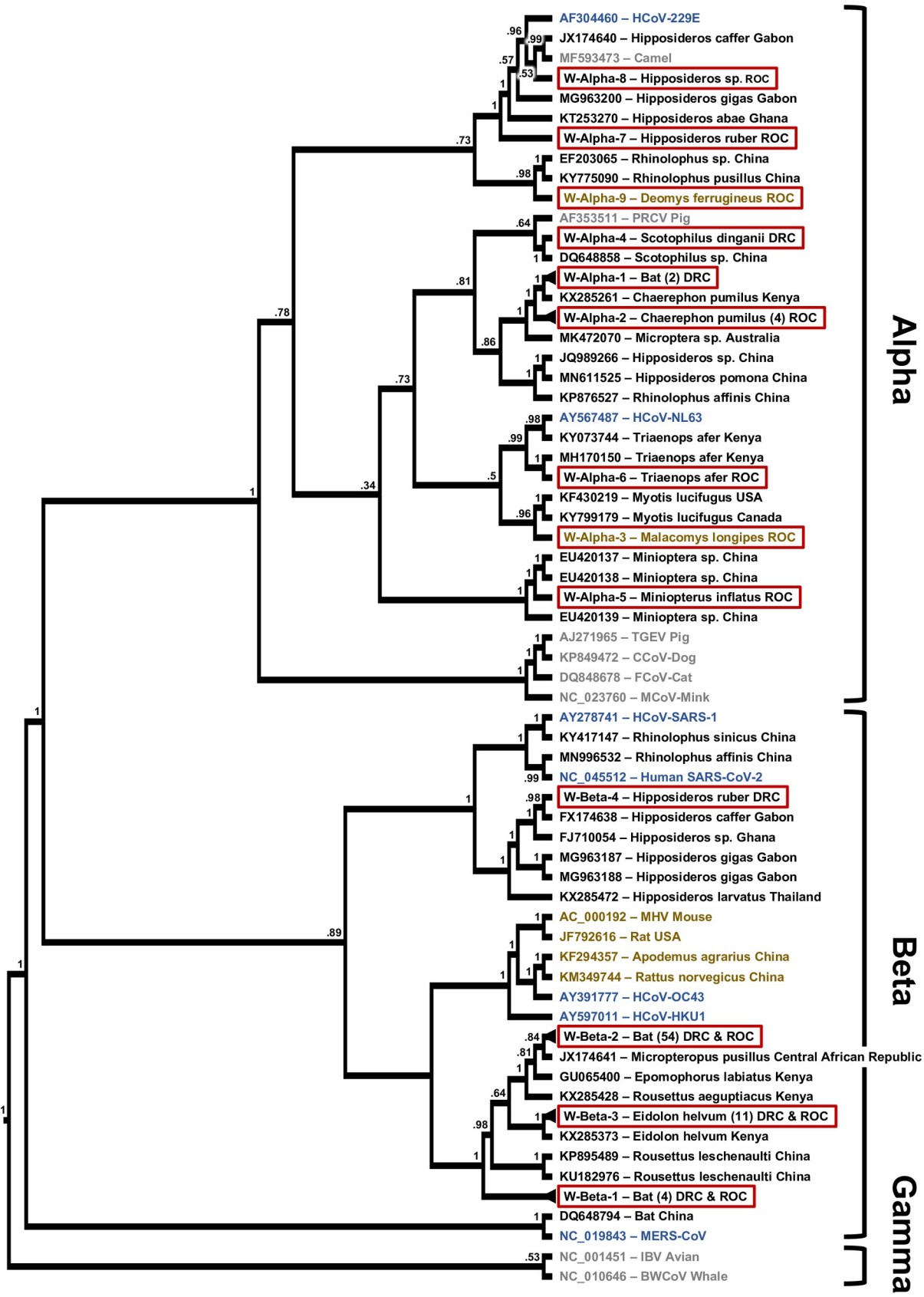

**Fig 2. Phylogenetic tree based on the RdRp region targeted by the PCR by Watanabe.** Maximum likelihood phylogenetic tree of coronavirus sequences presented as a proportional cladogram, based on the RdRp region targeted by the PCR by Watanabe et. al. [41]. The tree includes the sequences detected during the project (red boxes) and indicates the number of sequences sharing more than 95% nucleotide identities in brackets. GenBank accession numbers are listed for previously published sequences, while sequences obtained during the project are identified by cluster names (compare S2 Table). Black font indicates coronavirus sequences obtained from bats, brown font indicates rodents, blue humans and gray other hosts. The host species and country of sequence origin are indicated for bats and rodents if applicable. In case of clusters W-Alpha-1 sequences were detected in *Mops condylurus* and *Chaerephon sp.*, host species in cluster W-Beta-1 were *Megaloglossus woermanni* and *Epomops franqueti* and in case of cluster W-Beta-2 *Micropteropus pusillus*, *Epomops franqueti*, *Rhinolophus sp.*, *Myonycteris sp.*, *Mops condylurus*, *Megaloglossus woermanni*, and *Eidolon helvum* (compare S2 Table). Numbers at nodes indicate bootstrap support.

sequences were closely related to only two known viruses. Sequences with nucleotide identities of 97% or higher to Kenya bat coronavirus BtKY56 were found in 53 individual bats of 9 different species sampled on 14 occasions (Q-/W-Beta 2), while sequences with identities of 99% or higher to Eidolon bat coronavirus/Kenya/KY24 were detected in 30 individual bats of 3 different species sampled on 8 occasions (Q-/W-Beta 3) (S2 and S3 Tables). Bat coronavirus sequences in clusters Q-Alpha 1, 2, 7, and 8, W-Alpha 2 and 7, Q-/W-Beta 1, and Q-Beta 4 and 5 had identities of below 95% with known coronaviruses.

In three cases (Q-Alpha 1, W-Alpha 7 and 8), sequences were closest to coronaviruses found in bats and camels with a high similarity (>90% nucleotide identities) to human coronavirus 229E (Figs 2 and 3). Similarly, the viral sequences in clusters Q-Alpha 2 and Q-/W-Alpha 6 were most closely related to bat coronaviruses with some similarity (>80% nucleotide identities) to human coronavirus NL63 (Figs 2 and 3).

## Discussion

We detected coronavirus RNA in a significant proportion of the sampled bats (8.8%), but only in a small proportion of rodents (<1%) and none in NHPs or shrews. Finding relatively high numbers of coronavirus RNA positive bats is consistent with what has been previously reported; continuous high circulation of coronaviruses seems to be common especially in bats in tropical and subtropical climates [16, 32]. The specific PCR positive rates need to be approached with caution though, since factors such as species, season, location and others could play a role, as well as sample material and assays used for detection in comparison to other studies. Even within this study, as a result of its long duration and technological advances, some elements such as the collection medium or the brand of RNA extraction kits changed. Though minor in nature, we cannot completely rule out that they did have some kind of effect on the yield and thus detection rate. Our data suggest that intestinal/fecal samples might be best suited for screening, as all but one (20/21) single coronavirus in our data set were also and continuously more often detected in rectal swabs. This might be a result of prolonged shedding via feces compared to respiratory shedding. Only in one case did we detected coronavirus RNA in other tissue (spleen), which might indicate either viremia or an infection of the spleen itself, however a contamination with feces during necropsy cannot be ruled out with certainty.

Both assays used in the study were similarly effective in detecting both alpha- and betacoronavirus RNA in general, however, in most cases only one of the two was successful in amplifying viral RNA in a given sample. This indicates different sensitivities depending on the sequence (Subgenus/Species), and is to be expected given the nature of the degenerate primers. This confirms the benefits of using a combination of the two over using any of them alone for Coronavirus screening.

Our data suggest that coronavirus circulation in bats, at least in the Congo Basin, may indeed depend to some extent on species and seasonality (S2 Fig). We observed a significant

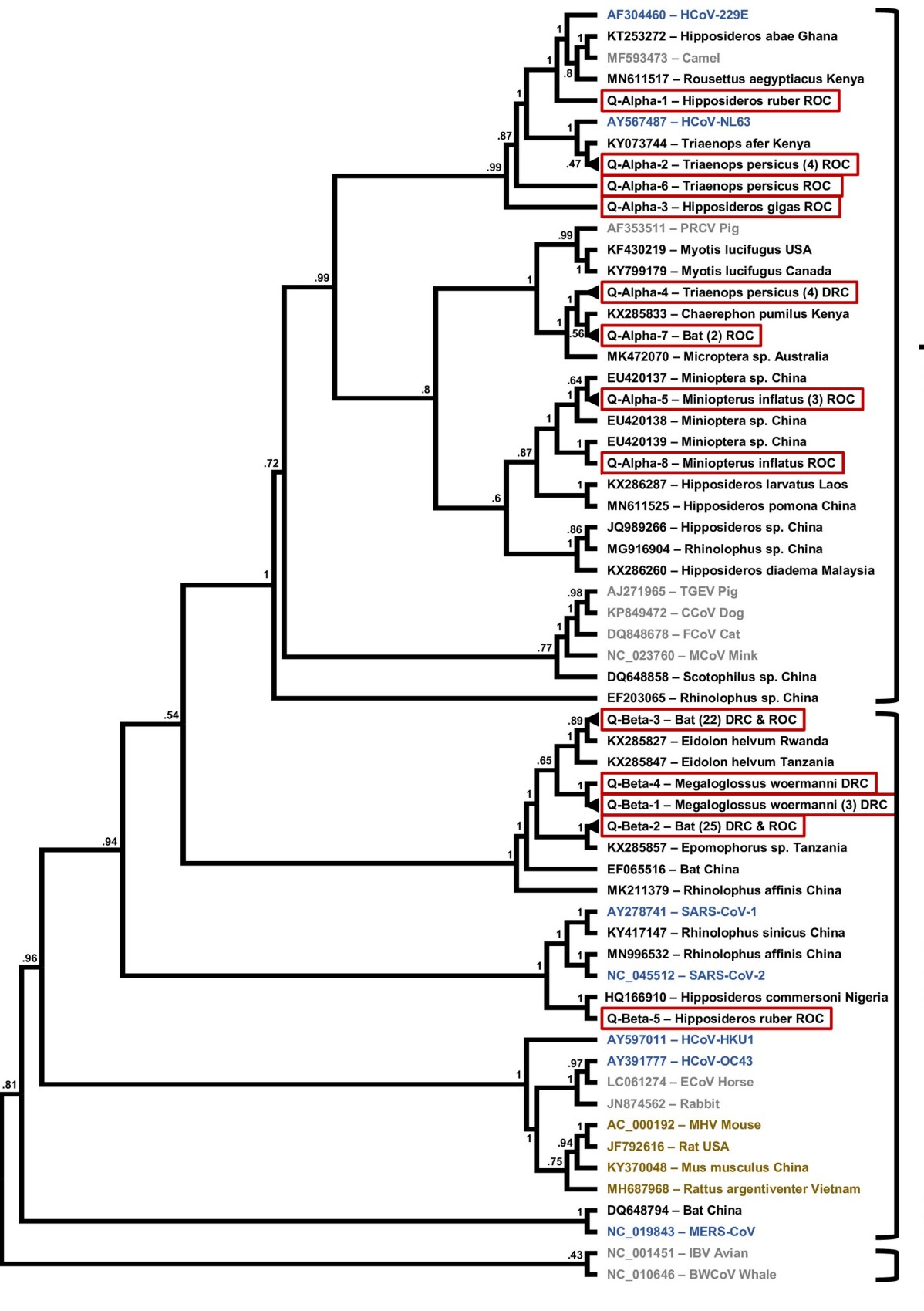

**Fig 3. Phylogenetic tree based on the RdRp region targeted by the PCR by Quan.** Maximum likelihood phylogenetic tree of coronavirus sequences presented as a proportional cladogram, based on the RdRp region targeted by the PCR by Quan et. al. [40]. The tree includes the sequences detected during the project (red boxes) and indicates the number of sequences sharing more than 95% nucleotide identities in brackets. GenBank accession numbers are listed for previously published sequences, while sequences obtained during the project are identified by cluster names (compare S1 Table). Black font indicates coronavirus sequences obtained from bats, brown font indicates rodents, blue humans and gray other hosts. The host species and country of sequence origin are indicated for bats and rodents if applicable. In case of clusters Q-Alpha-4 sequences were detected in *Mops condylurus* and *Chaerephon sp.*, host species in cluster Q-Alpha-7 were *Epomops franqueti* and *Chaerephon pumilus*, in case of cluster Q-Beta-2 *Micropteropus pusillus* and *Epomops franqueti*, and for cluster Q-Beta-3 *Megaloglossus woermanni*, *Eidolon helvum*, and *Epomops franqueti* (compare S2 Table). Numbers at nodes indicate bootstrap support.

difference in the number of bats testing positive depending on the local calendric season (p = 0.0176), with 10.5% of coronavirus RNA positive bats in the wet season but only 6.6% in the dry season at similar sample sizes for both seasons (Table 1). Interestingly, when looking at the family and species level, this holds true only for the *Pteropodidae* and *Rhinolophidae* species (p < 0.0001) while *Hipposideridae*, *Miniopteridae*, *Molossidae*, and *Vespertilionidae* species are more likely to be positive for coronavirus RNA in the dry season (p < 0.0001) (Table 1). The latter, though not for those specific families but for bats in general, has been proposed to be the correlation on a global scale [16]. We can only speculate as to the reasons of the apparent seasonality, but family and species seem to be important determinants. The birthing season of a species, which is often dependent on the local characteristics of the calendric season, has been suggested as a determinant for coronavirus spikes in bat populations, as juvenile bats become susceptible to infection once maternal antibody levels wane [32, 39]. Due to the diverse set of species in our sample set, individual sample numbers for most species are too small to draw definite conclusions, however the significant seasonal difference between *Yinpteorchiroptera* and *Yangochiroptera* are largely supported by respective trends in the individual species. We tested if the results from any particular species might be responsible for the observed correlation between season and the rate of positive coronavirus RNA animals. The only species that turned out to have a strong influence on the outcome was *Eidolon helvum*. However, the effect of dropping it from the analysis did only influence the outcome for bats in total, while it was not strong enough to negate the observed statistical significances for season within the *Pteropodidae* family or the *Yinpterochiroptera* suborder.

We did find *Eidolon helvum*, a bat usually roosting in large colonies, to be significantly overrepresented among the coronavirus positive bats (p = 0.0005), and higher detection rates in this species have been reported before [16, 44]. However, samples from *Eidolon helvum* in this study were collected from animals sold at two different markets on seven different days, and although we detected coronavirus RNA in some *Eidolon helvum* bats obtained at each of those occasions, it is possible that many of these bats came from the same roosts. The fact that all but one of the *Eidolon helvum* bats were found to be positive for the same coronavirus type (Q-/W-Beta-3) supports the assertion that there may be a connection between those bats. Our dataset does contain evidence that bat coronaviruses are readily shared within local bat populations. In fact, 109 out of 119 coronavirus positive bats were from sampling events with at least one other coronavirus RNA positive bat, and in all but six of these cases there was another bat with the same coronavirus type in the event-cohort (S2 and S3 Tables). Even though we cannot pinpoint the exact roosting relationship between all of these bats, this does confirm that coronaviruses are readily shared among the bats in an area, even across species boundaries. It also highlights that several different coronaviruses can circulate in parallel, including occasional double infections (S2 Table).

We found a much higher percentage of bats that were part of the bushmeat value chain to be positive for coronavirus RNA, which could have significant implications for the risk of coronavirus spillover from bats into humans. In our data set, 81% of the value chain samples

were collected in the wet season, and this group also contained all of the *Eidolon helvum* samples. This suggests that seasonality and preferentially hunted species (80% *Pteropodidae*) are likely responsible for the higher rate of coronavirus RNA positive animals in the value chain. According to our data, it also seems that male animals are overrepresented among bats with coronavirus positive samples. It is possible that behavioral differences between males and females play a role, such as reduced activity of females during the time of birthing and breast-feeding or higher stress levels among males during the breeding season [45]. Further investigation is required to confirm and assess the reasons for this observation.

It appears clear from our findings, that bats rather than rodents or primates are sustaining a significant circulation of coronaviruses in the Congo Basin. Evidence for coronavirus circulation in wild animals other than bats is generally much scarcer, even though civets, raccoon, dogs, and camels have been shown to be involved in outbreaks of SARS and MERS [8, 11, 16].

We estimate that the 121 detected sequences correspond to 21 different coronaviruses based on the differences between the amplified sequences, considering the conserved nature of the amplified fragments within the RdRp open reading frame (ORF). These 21 coronaviruses include some that appear to only be distantly related to already described coronaviruses, and others that have already been found elsewhere, such as Kenya bat coronavirus BtKY56 and Eidolon bat coronavirus/Kenya/KY24 (Figs 2 and 3, S2 Table). In several instance the coronaviruses detected here are closely related to ones found in neighboring countries such as Gabon and the Central African Republic as well, which could be expected, as the political borders in Central Africa largely do not coincide with natural barriers that would prevent bats from crossing (Figs 2 and 3).

RNA of either Kenya bat coronavirus BtKY56 or Eidolon bat coronavirus/Kenya/KY24 was detected in ~70% (83) of the positive bats in this study and in several hundred bats reported previously (GenBank). Interestingly Kenya bat coronavirus BtKY56 appears to be a common virus species in the Congo Basin, while elsewhere it appears to be Eidolon bat coronavirus/Kenya/KY24 that is more common. These observations are undoubtedly susceptible to a sampling bias, for example due to the species composition of sample sets, particularly with *Eidolon helvum*, which can be sampled in large numbers when colonies are present or when they are present in markets [46]. A study conducted in neighboring Congo Basin country Gabon, for which *Miniopterus*, *Rousettus*, *Hipposideros*, *Macronycteris*, and *Coleura* bats were sampled found primarily RNA closely related to human coronaviruses 229E [32]. However, we do find evidence of Kenya bat coronavirus BtKY56 and Eidolon bat coronavirus/Kenya/KY24 in a relative wide array of bat hosts, indicating that species barriers may not be a limiting factor for sharing these specific betacoronaviruses (Figs 2 and 3, S2 Table). In contrast, most of the other sequences that we detected with related sequences in GenBank were detected in bats of the same genus by us and previously by others, supporting some degree of general species specificity and virus host co-evolution despite the latent ability of at least some coronaviruses to jump species barriers within and outside of the taxonomic order of hosts [15, 16, 18]. How often these events occur is not fully understood, but it is generally assumed that bats serve as a reservoir for coronaviruses [17]. With SARS-CoV-1 and SARS-CoV-2, the available evidence suggests that they were successfully transmitted from bats into humans, either directly or indirectly [21]. When we add to these two coronaviruses MERS that originated in bats and established a sustained reservoir in camels with occasional spillover into humans, we have witnessed three coronavirus spillover events with a bat origin in less than two decades. Considering our increased awareness and abilities to detect the emergence of novel viruses, it can be assumed that there may have been multiple coronavirus zoonotic events in the past that either led to some degree of either self-limiting outbreaks, or may have established a permanent virus host relationship with a new host [47]. MERS-CoV and SARS-CoV-1 may represent examples

for the former, while the latter may be represented by human coronaviruses 229E and NL63; the ultimate outcome with regards to SARS-CoV-2 remains undetermined however. In our study we also detected viral RNA related to human coronaviruses 229E and NL63 in eight bats, and in a recent study from Gabon RNA closely related to human coronaviruses 229E was found in 12 out of 18 coronavirus RNA positive bat samples [32]. Whether or not these relatives of human pathogens or other strains of the coronaviruses currently circulating in bats can and will jump into humans in the future is difficult to predict at present. Progress in the understanding of molecular processes such as RNA polymerase proofreading capability, receptor usage, as well as in the field of human behavior are however certainly helping our understanding of risk [48]. The close contact of humans with wildlife including bats in the Congo Basin, especially in the context of hunting and wild animal trade, are certainly factors contributing to a higher risk for zoonotic events involving coronaviruses or other infectious agents.

The two sequences we detected in rodents (*Deomys ferrugineus* and *Malacomys longipes*) likely correspond to novel alphacoronaviruses. The lack of sequences closely related to the two indicates that rodent coronaviruses may be an understudied field, especially considering that rodents are the largest family of mammals.

We conclude overall, that bats and to a much smaller degree rodents in the Congo Basin harbor diverse coronaviruses, of which some might have the molecular potential for spillover into humans. Considering the close contact between wildlife and humans in the region, as part of the value chain or in peri-domestic settings, there is an elevated and potentially increasing risk for zoonotic events involving coronaviruses. Thus, continued work to understand the diversity, distribution, molecular mechanisms, host ecology, as well as consistent surveillance of coronaviruses at likely hotspots, are critical to help prevent future global pandemics.

## Supporting information

**S1 Fig. Sample sites around the cities of Brazzaville and Kinshasa.** Geographical map indicating all sampling sites in and around the urban centers of Brazzaville and Kinshasa on either side of the Congo river, the border between the Republic of Congo and the Democratic Republic of the Congo. Locations where coronavirus RNA was detected in bats are highlighted with blue triangles, sampling sites without viral RNA detection are marked by black dots. Base map and data from OpenStreetMap and OpenStreetMap Foundation.
(TIFF)

**S2 Fig. Bat sampling effort and detection rates by calendar month.** Bat sampling effort and detection rates by calendar month (cumulative over all years). Panel A is depicting the percentage of the total samples collected in each month, relative to total bat samples collected. Panel B depicts the percentage of coronavirus RNA positive animals relative to each month's total samples, while panel C shows the percentage of sampling events with at least one coronavirus RNA detection per month. Blue bars are indicating that all samples were collected during local rainy season, while beige indicates the same for the local dry season. Gradient colored bars indicate months in which dry and wet season samples were collected depending on the location.
(PDF)

**S1 Table. Samples collected.**
(XLS)

**S2 Table. Coronavirus RNA positive samples.**
(XLSX)

**S3 Table. Geographical and temporal distance between sampling events.** Geographical and temporal distance between sampling events with detections of identical or closely related (>95% nucleotide identities) coronavirus RNA sequences (compare also S1 and S2 Tables). (DOCX)

## Acknowledgments

The authors would like to thank: The governments of the Democratic Republic of the Congo and of the Republic of Congo for the permission to conduct this study; the staff of Metabiota, Wildlife Conservation Society, and EcoHealth Alliance who assisted in sample collection and testing and other members of the PREDICT-1 and PREDICT-2 consortium (https://ohi. vetmed.ucdavis.edu/programs-projects/predict-project/authorship).

## Author Contributions

**Conceptualization:** Nathan D. Wolfe, Christian E. Lange.

**Data curation:** Charles Kumakamba, Placide Mbala Kingebeni, Maria Makuwa, Amethyst Gillis, Matthew LeBreton, Sarah H. Olson, Kenneth Cameron, David J. McIver, James A. Ayukekbong, Tracey Goldstein, Christian E. Lange.

**Formal analysis:** Charles Kumakamba, Matthew LeBreton, Damien O. Joly, Christian E. Lange.

**Funding acquisition:** Ubald Tamufe, Tracey Goldstein, Damien O. Joly, Nathan D. Wolfe, Jean J. Muyembe Tamfum.

**Investigation:** Charles Kumakamba, Fabien R. Niama, Francisca Muyembe, Jean-Vivien Mombouli, Placide Mbala Kingebeni, Rock Aime Nina, Ipos Ngay Lukusa, Gerard Bounga, Frida N'Kawa, Joseph Atibu Losoma, Maria Makuwa, Amethyst Gillis, Alex Tremeau-Bravard, Brett R. Smith, Jasmine Pante, Anne W. Rimoin, Damien O. Joly, Karen Saylors, Christian E. Lange.

**Methodology:** Matthew LeBreton, Anne W. Rimoin, Tracey Goldstein, Nathan D. Wolfe.

**Project administration:** Charles Kumakamba, Fabien R. Niama, Cynthia Goma Nkoua, Prime Mulembakani, Maria Makuwa, Ubald Tamufe, Amethyst Gillis, Sarah H. Olson, Kenneth Cameron, Patricia Reed, Alain Ondzie, Bradley S. Schneider, David J. McIver, James A. Ayukekbong, Nicole A. Hoff, Anne W. Rimoin, Anne Laudisoit, Corina Monagin, Tracey Goldstein, Damien O. Joly, Karen Saylors, Nathan D. Wolfe, Edward M. Rubin, Romain Bagamboula MPassi, Jean J. Muyembe Tamfum.

**Resources:** Corina Monagin, Tracey Goldstein, Karen Saylors, Edward M. Rubin, Jean J. Muyembe Tamfum.

**Software:** Damien O. Joly.

**Supervision:** Charles Kumakamba, Fabien R. Niama, Jean-Vivien Mombouli, Placide Mbala Kingebeni, Cynthia Goma Nkoua, Prime Mulembakani, Maria Makuwa, Amethyst Gillis, Matthew LeBreton, Sarah H. Olson, Kenneth Cameron, Patricia Reed, Bradley S. Schneider, James A. Ayukekbong, Nicole A. Hoff, Anne W. Rimoin, Anne Laudisoit, Corina Monagin, Tracey Goldstein, Damien O. Joly, Karen Saylors, Nathan D. Wolfe, Edward M. Rubin, Romain Bagamboula MPassi, Jean J. Muyembe Tamfum.

**Validation:** Sarah H. Olson, Damien O. Joly.

**Visualization:** David J. McIver, Christian E. Lange.

**Writing – original draft:** Charles Kumakamba, Amethyst Gillis, Christian E. Lange.

**Writing – review & editing:** Charles Kumakamba, Fabien R. Niama, Francisca Muyembe, Jean-Vivien Mombouli, Placide Mbala Kingebeni, Rock Aime Nina, Ipos Ngay Lukusa, Gerard Bounga, Frida N'Kawa, Cynthia Goma Nkoua, Joseph Atibu Losoma, Prime Mulembakani, Maria Makuwa, Ubald Tamufe, Amethyst Gillis, Matthew LeBreton, Sarah H. Olson, Kenneth Cameron, Patricia Reed, Alain Ondzie, Alex Tremeau-Bravard, Brett R. Smith, Jasmine Pante, Bradley S. Schneider, David J. McIver, James A. Ayukekbong, Nicole A. Hoff, Anne W. Rimoin, Anne Laudisoit, Corina Monagin, Tracey Goldstein, Damien O. Joly, Karen Saylors, Nathan D. Wolfe, Edward M. Rubin, Romain Bagamboula MPassi, Jean J. Muyembe Tamfum, Christian E. Lange.

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
