## [Decision Letter · Decision Letter 0]

31 Dec 2020

PONE-D-20-20865

Coronavirus surveillance in Congo basin wildlife detects RNA of multiple species circulating in bats and rodents

PLOS ONE

Dear Dr. Lange,

Thank you for submitting your manuscript to PLOS ONE. Our apologies for the extensive review period. We had major difficulties in assigning reviewers and obtaining the reviews in time. After careful consideration, we feel that it has merit but does not fully meet PLOS ONE’s publication criteria as it currently stands. Therefore, we invite you to submit a revised version of the manuscript that addresses the points raised during the review process.

Please address all comments indicated by reviewer 1 and 2 especially more detail about the methodology and including additional sequences from the geographical area in the analysis and discussions. I do appreciate that obtaining larger gene regions are complicated but if at all possible, please include this. 

We look forward to receiving your revised manuscript.

Kind regards,

Wanda Markotter

Academic Editor

PLOS ONE

2. In your Methods section, please provide additional location information of the collection sites, including geographic coordinates for the data set if available.

4. We note that Figure 1 and Supplement_1 file in your submission contain map  images which may be copyrighted. All PLOS content is published under the Creative Commons Attribution License (CC BY 4.0), which means that the manuscript, images, and Supporting Information files will be freely available online, and any third party is permitted to access, download, copy, distribute, and use these materials in any way, even commercially, with proper attribution. For these reasons, we cannot publish previously copyrighted maps or satellite images created using proprietary data, such as Google software (Google Maps, Street View, and Earth). For more information, see our copyright guidelines: http://journals.plos.org/plosone/s/licenses-and-copyright.

(1) You may seek permission from the original copyright holder of Figure 1 and Supplement_1 file to publish the content specifically under the CC BY 4.0 license. 

6. Thank you for stating the following in the Financial Disclosure section:

"NW and TG received awards GHN-A-OO-09-00010-00 and AID-OAA-A-14-00102 from the United States Agency for International Development (https://www.usaid.gov). The funders had no role in study design, data collection and analysis, decision to publish, or preparation of the manuscript."

We note that one or more of the authors are employed by a commercial company: 'Metabiota Inc', 'Labyringth Global Health', 'Development Alternatives, Inc.', 'Mosaic, Cameroon', 'Etiologic, Oakland' and 'Pinpoint Science, San Francisco'.

(2) Please also provide an updated Competing Interests Statement declaring this commercial affiliation along with any other relevant declarations relating to employment, consultancy, patents, products in development, or marketed products, etc.  

Reviewers' comments:

Reviewer's Responses to Questions

**Comments to the Author**

1. Is the manuscript technically sound, and do the data support the conclusions?

Reviewer #1: Yes

Reviewer #2: Partly

2. Has the statistical analysis been performed appropriately and rigorously? 

Reviewer #1: Yes

Reviewer #2: Yes

3. Have the authors made all data underlying the findings in their manuscript fully available?

Reviewer #1: Yes

Reviewer #2: Yes

4. Is the manuscript presented in an intelligible fashion and written in standard English?

Reviewer #1: Yes

Reviewer #2: Yes

5. Review Comments to the Author

Reviewer #1: The paper reports on the surveillance of coronaviruses among bats, rodents, shrews and non-human primates in the congo basin. The study was performed across 12 years and opportunistically sampled 3561 animals for coronavirus RNA using two separate assays. Nearly all of the positives were from bats with the additional detection of two rodent positives. Based on the results the authors also performed some analyses to correlate detection of positives with observed ecological features such as seasonality. The paper is well written and the conclusions are thoughtful and well-reasoned.

My main request is the need to better details on the sampling methodology of the animals sampled- specifically in terms of time and frequency of sampling and repeat sampling at the same site if performed. This can be generally included in the text or better illustrated in one of the figures. The authors report that they identified no coronaviruses from non-human primates or shrews and only two from rodents. They comment that CoVs in wildlife are scarcer than in bats, and this may be true but consider the fragmented opportunistic sampling generally dedicated to sampling of “non-bats” for coronaviruses. It is difficult to discern from the way the data is reported in the manuscript or even from the supplementary table 2 how dedicated or opportunistic the sampling has been for the non-human primates across the 836 individuals from the total sampling timespan. Were some of the animals sampled more than once? If so, it would be important to highlight in text (that dedicated and repeat sampling was conducted).

My second concern is regarding the two assays used in manuscript. Supp 2 indicates a real time assay used for some samples – though this was not indicated in text- can the authors please add or clarify this? Also can the authors please clarify if the Quan assay was modified from the original? were the primers optimized? The Watanabe assay was modified as indicated. Can the authors comment on the sensitivity between the assays? Or what controls were used for the assays? The reason for my query is that these assays don’t amplify the same gene region of the genome (of course) and so the sequences cannot be directly compared so it might be a little difficult to determine if the two assays identified sequences from the same or different viruses. It seems that it was somewhat frequent that one assay was capable of detecting sequences missed by the other assay. What was the frequency of overlap that the assays were capable of detecting the same ‘virus’ vs. the frequency that the assays missed or detected very different viruses? Then, did the authors try combining the two assays on the positives to amplify the sequence region between primer regions in order to amplify both a longer gene region but also confirm identification of the same viral sequences with the different assays? Is one assay better at detecting a specific clade of coronavirus? As this was the first report of the rodent coronaviruses, full genomes would have been more informative.

I have no specific line issues except that the authors can consider using ‘alphacoronaviruses’ or ‘betacoronaviruses’ instead of Alpha coronaviruses or Beta coronaviruses (line 334 and 308).

Can the authors define the term wet season? Dry season? Exactly which months does this include? Similarly, in supplement 4 – which months are indicated by 1-12? Which years?

Its difficult to determine how the sequences were named in the study – please consider the naming convention suggested by the Coronavirus study group: The species Severe acute respiratory syndrome related coronavirus: classifying 2019-nCoV and naming it SARS-CoV-2 (page 4). Using a method that names sequences based on most related sequences as has previously been done creates significant confusion.

Please consider adding sequence identifiers to the sequences in the phylogenies? Like HKU8 to bat sequence Miniopterus from China etc. It also helps to orient the reader in the tree and relate it to previously reported coronaviruses (and species). Moreover, it is challenging to navigate between the trees so determine if similar coronaviruses were detected with the different assays, as the same reference sequences are not present in all trees- is this due to the regions availability to compare? (particularly between the 229E and NL63 clades).

Reviewer #2: Manuscript Number PONE-D-20-20865

Reviewer's Comments

General Comment :

The study provides new information on the diversity of coronavirus species circulating in the Democratic Republic of Congo and the Republic of Congo. However, at any time, this study does not refer to a recent published study on coronaviruses conducted in Gabon, a country in the Congo Basin.

In addition, full-length genomes with particularly spike gene at least will increase the importance of the work.

Specific comments :

Page 1, lines 1-2 : As the Congo Basin is not limited to these two countries, I propose that the title be more precise. For example « Coronavirus surveillance in two countries in Congo basin wildlife detects … ».

Page 4, lines 91-94 : The authors could also refer to the study by Maganga et al. (Maganga GD, Pinto A, Mombo IM, et al. 2020. Scientific Reports. 10:7314, 10.1038/s41598-020-64159-1) carried out in a Congo Basin country.

Page 5

Line 102 : To determine the degree of coronavirus circulation, a serological study would be more appropriate, therefore I suggest this sentence : « Our goal was to determine the diversity of circulating coronaviruses, using… ».

Line 111 : The authors state « Fecal samples were collected from free-ranging NHPs ». Can you explain how the collection was done?

Pages 5-6

Lines 120-121 : Why extraction with Trizol in addition to extraction using commercial kits? And why different extraction kits for swabs collected prior to and after 2014?

Lines 124-138 : The authors used two different PCR assays that amplify the same gene (RdRp), to broaden the detection spectrum of coronaviruses regardless of the animal host. However, the sizes of the amplified fragments are quite short for a highly conserved gene such as RdRp. It would be relevant to obtain sequences of a gene of interest such as the Spike gene if complete genomes cannot be obtained.

Page 7

Line 172 : The collection period should be indicated in the Materials and Methods section.

Line 177 : The authors state “others (36)“ as tissue sample. Can you specify the nature of these samples ?

Lines 178-179 : Authors should be careful not to create confusion between samples and animals. The authors indicate that 121 animals were detected positive, corresponding to 23 oral swabs, 102 rectal swabs and 1 pool of liver and spleen. However, for the 83 and 73 samples in which viral RNA was found, can we know how many individuals these numbers correspond to?

Were the 73 samples that tested positive using Quan PCR assay the same as the 83 samples that tested positive using Watanabe PCR assay ? If so, this should be specified.

Can the authors explain why the search for coronaviruses was done in the liver and spleen? These are not organs that are conventionally screened for these viruses.

Page 8

Lines 187-190 : As was done for rodents, indicate for bats, in brackets, the number of positive individuals out of the total number tested.

At any time in the materials and methods section there was no mention of how the identification of the species was done.

Lines 194-195 : In the introduction, I suggest that the classification of bats be briefly presented so that the terms Yinpterochiroptera and Yangochiroptera do not appear abruptly in the results section. Also, write throughout the document Yinpterochiroptera instead of Yinpterchiroptera. And write them in italics throughout the document.

Page 10

Lines 243-246 : The authors should discuss further the seasonal difference in positivity rates. For example, what factors related to bats might explain them? Maganga et al. 2020 had, for the species studied, identified the rainy season as the birthing season, which may justify a higher viral shedding rate.

Page 22

Table 1 : Correct Yinpterochiroptera

Figures

Figure 2 and 3 : I suggest the authors to add sequences of Alphacoronavirus of bats from Gabon and to include some sequences of Betacoronavirus of bats from this area. Especially since Figure 2 seems to show that the same viruses would circulate in Gabon and the Republic of Congo (See the first cluster of the phylogenetic tree). Moreover, the authors do not even discuss this observation.

6. PLOS authors have the option to publish the peer review history of their article (what does this mean?). If published, this will include your full peer review and any attached files.

Reviewer #1: No

Reviewer #2: No

---

## [Author Response · Author response to Decision Letter 0]

10 Apr 2021

Included as word document "Response_to_Reviewers".

---

## [Decision Letter · Decision Letter 1]

20 May 2021

Coronavirus surveillance in wildlife from two Congo basin countries detects RNA of multiple species circulating in bats and rodents

PONE-D-20-20865R1

Dear Dr. Lange,

We’re pleased to inform you that your manuscript has been judged scientifically suitable for publication and will be formally accepted for publication once it meets all outstanding technical requirements.

Kind regards,

Wanda Markotter

Academic Editor

PLOS ONE

Additional Editor Comments (optional):

Reviewers' comments:

Reviewer's Responses to Questions

**Comments to the Author**

1. If the authors have adequately addressed your comments raised in a previous round of review and you feel that this manuscript is now acceptable for publication, you may indicate that here to bypass the “Comments to the Author” section, enter your conflict of interest statement in the “Confidential to Editor” section, and submit your "Accept" recommendation.

Reviewer #1: All comments have been addressed

Reviewer #2: All comments have been addressed

2. Is the manuscript technically sound, and do the data support the conclusions?

Reviewer #1: Yes

Reviewer #2: (No Response)

3. Has the statistical analysis been performed appropriately and rigorously? 

Reviewer #1: Yes

Reviewer #2: (No Response)

4. Have the authors made all data underlying the findings in their manuscript fully available?

Reviewer #1: Yes

Reviewer #2: (No Response)

5. Is the manuscript presented in an intelligible fashion and written in standard English?

Reviewer #1: Yes

Reviewer #2: (No Response)

6. Review Comments to the Author

Reviewer #1: The authors have effectively updated the manuscript to resolve issues raised by the first draft. Particularly the material and methods and results sections are much more informative and contribute interesting information.

I recommend approval of the manuscript but urge the authors to please review their host bat taxonomy as this is an expanding field (to avoid mistakes and conflicts with the manuscript in future). The species Trianeops persicus has been restricted to the middle East and according to several sources (Benda and Vallo 2009 and The African Chiropteran Report) the only species in Africa is T. afer. Similarly molossid bats have undergone taxonomic changes replacing Chaerephon with Mops, within the Pteropodidae (changes to Micropusillus), and many changes to the genera in the Vespertilionidae family (particularly to Neoromicia, Laephotis etc (Monadjem et al. 2020).

Reviewer #2: (No Response)

7. PLOS authors have the option to publish the peer review history of their article (what does this mean?). If published, this will include your full peer review and any attached files.

Reviewer #1: No

Reviewer #2: No

---

## [Editor Report · Acceptance letter]

3 Jun 2021

PONE-D-20-20865R1 

Coronavirus surveillance in wildlife from two Congo basin countries detects RNA of multiple species circulating in bats and rodents 

Dear Dr. Lange:

I'm pleased to inform you that your manuscript has been deemed suitable for publication in PLOS ONE. Congratulations! Your manuscript is now with our production department. 

Kind regards, 

on behalf of

Prof Wanda Markotter 

Academic Editor

PLOS ONE